# The Dual-Targeted Fusion Inhibitor Clofazimine Binds to the S2 Segment of the SARS-CoV-2 Spike Protein

**DOI:** 10.3390/v16040640

**Published:** 2024-04-20

**Authors:** Matthew R. Freidel, Pratiti A. Vakhariya, Shalinder K. Sardarni, Roger S. Armen

**Affiliations:** Department of Pharmaceutical Sciences, College of Pharmacy, Thomas Jefferson University, 901 Walnut St. Suite 918, Philadelphia, PA 19170, USApratitiashish.vakhariya@students.jefferson.edu (P.A.V.); shalindersardarni1998@gmail.com (S.K.S.)

**Keywords:** Clofazimine, Arbidol, Toremifene, fusion inhibitor, Spike-dependent, SARS-CoV-2, S2 segment, S2 subunit, Nsp13 helicase, surface plasmon resonance, molecular docking, CHARMM

## Abstract

Clofazimine and Arbidol have both been reported to be effective in vitro SARS-CoV-2 fusion inhibitors. Both are promising drugs that have been repurposed for the treatment of COVID-19 and have been used in several previous and ongoing clinical trials. Small-molecule bindings to expressed constructs of the trimeric S2 segment of Spike and the full-length SARS-CoV-2 Spike protein were measured using a Surface Plasmon Resonance (SPR) binding assay. We demonstrate that Clofazimine, Toremifene, Arbidol and its derivatives bind to the S2 segment of the Spike protein. Clofazimine provided the most reliable and highest-quality SPR data for binding with S2 over the conditions explored. A molecular docking approach was used to identify the most favorable binding sites on the S2 segment in the prefusion conformation, highlighting two possible small-molecule binding sites for fusion inhibitors. Results related to molecular docking and modeling of the structure–activity relationship (SAR) of a newly reported series of Clofazimine derivatives support the proposed Clofazimine binding site on the S2 segment. When the proposed Clofazimine binding site is superimposed with other experimentally determined coronavirus structures in structure–sequence alignments, the changes in sequence and structure may rationalize the broad-spectrum antiviral activity of Clofazimine in closely related coronaviruses such as SARS-CoV, MERS, hCoV-229E, and hCoV-OC43.

## 1. Introduction

Early in the COVID-19 pandemic, both Arbidol **1** [1,2], and Clofazimine **2** [2,3,4,5,6,7,8,9,10], were identified to be clinical drugs that were effective in vitro inhibitors of SARS-CoV-2, according to drug-repurposing screens. Clofazimine was found in at least eight independent drug-repurposing efforts [2,3,4,5,6,7,8,9,10] and was commonly identified among the most promising compounds in those studies based on screening and antiviral assay data [2,3,4,5,6,7,8,9,10]. Drug-repurposing efforts are important to identify possible drugs available for clinical use when patients are not able to receive “standard-of-care” drug treatment [11,12]. In addition to this, ongoing studies on effective direct-acting antivirals [13,14] are important to understand the direct mechanism of action and the molecular basis for broad-spectrum activity. Both Arbidol and Clofazimine, shown in Figure 1, have been reported to act as SARS-CoV-2 viral fusion inhibitors [1,5,6,9] and have demonstrated synergistic antiviral activity with the Nsp12 inhibitor Remdesivir [4,5,15]. Small-molecule fusion inhibitors [14] with this profile are attractive, as they may be expected to exhibit antiviral activity with either Nsp12 replication inhibitors such as Remdesivir (Veklury) [16] or Nsp5 Main Protease (MPro) inhibitors such as nirmatrelvir (Paxlovid) [17,18].

Our previous studies focused on the mode of action of Arbidol as a fusion inhibitor [19,20]. These studies showed that it was only a partial inhibitor of SARS-CoV-2 in cytopathic effect (CPE) assays [20], in comparison to full inhibitors such as Remdesivir. An early drug-repurposing screen using pseudotyped virus particles, conducted by Chen et al., reported similar results, where a range of viral entry inhibitors, including NKH477 and trimipramine, were found to only be partial inhibitors in similar SARS-CoV-2 infection CPE assays in Vero E6 cells [21]. In comparison, several studies have shown that Clofazimine is a full inhibitor of SARS-CoV-2 in a range of infection assay cell types (Vero E6, Huh7, and Caco-2 cells) [4,5,6] including physiologically relevant cell lines (cardiomyocytes, Calu-3 and human primary airway epithelial cells) [5,8]. In addition to this, **2** has also shown impressive preclinical antiviral activity in a golden Syrian hamster animal model [5]. As a clinical repurposing drug, Clofazimine is a drug with clinical drawbacks and is associated with numerous adverse side effects for systemic use (abdominal pain, gastrointestinal tract disturbances) [22]. Some specific adverse effects (skin discoloration and tissue accumulation) are directly due to the physiochemical properties of the drug due to its high lipophilicity (cLogP = 7.1), which results in accumulation in various lipophilic tissues. Of course, it is important to consider clinical factors that extend beyond the potential utility of the drug. Clofazimine is generally well tolerated, though in some patient populations it may not be an ideal choice, as it can cause skin discoloration that may persist for months or even years after the completion of therapy. Clofazimine may also not be an appropriate medication in patients taking medications that prolong the QTc interval, as it can increase the risk of potentially fatal arrhythmias such as Torsade de Pointes, and a physician or pharmacist should review the patient’s medications prior to administering this treatment due to the possibility of drug interactions.

Due to these specific adverse effects, recent synthetic medicinal chemistry efforts have been aimed at generating improved derivatives of Clofazimine with lower (cLogP) values and greater solubility to reduce these side effects [23]. However, despite numerous adverse effects, scientists worldwide have a great deal of clinical experience using Clofazimine to treat various forms of leprosy and drug-resistant tuberculosis, and the systemic toxicity can be significantly reduced by delivering the drug through inhalation [24,25]. Clofazimine has also been considered an experimental drug for other infections and is on the World Health Organization’s List of Essential Medicines [26]. Clofazimine remains an important clinical drug that may be considered for the treatment of COVID-19 [27], and its relatively low cost compared to other recently approved small-molecule inhibitors or antibody-based therapies serves to improve access to care for needy populations. While the fusion inhibitor activity of Clofazimine has been shown to be Spike-dependent [5,23], to our knowledge, the precise mechanism of action by which it acts as a fusion inhibitor has yet to be determined. In this paper, we demonstrate that Clofazimine binds to the S2 segment of Spike using a Surface Plasmon Resonance (SPR) binding assay [28,29], which helps to delineate how it acts as a potent fusion inhibitor.

An overview of the SPR direct binding assay setup (Figure 2A) shows how a full-length Spike protein construct may be immobilized on a SPR biosensor flow cell surface (FC2) and tested against a reference cell (FC1). A purified construct of the Spike S2 segment is immobilized on (FC4) and tested against a reference cell (FC3). This assay design allows simultaneous determination of small-molecule binding to both the full-length Spike protein and the S2 segment. Shown in Figure 2A are four sample surfaces of the biosensor chip corresponding to flow cells (FC1 to FC4) where protein samples may be immobilized to measure small-molecule binding. During a binding experiment, the reference subtracted signal of FC2-1 and FC4-3 provides simultaneous measurement of binding to the full-length Spike and the S2 segment.

We demonstrate that Arbidol and Clofazimine bind to the S2 segment of Spike. As molecular docking and structure–activity relationship (SAR) modeling work from our laboratory successfully predicted the binding site of Arbidol on the S2 segment of Spike [19,20], as had been achieved previously by Vankadari with docking alone [30], another publication from Shuster et al. [31] experimentally confirmed the binding site of Arbidol on the S2 segment of Spike, as shown in Figure 3A,C. Elegant work from Shuster et al. [31] independently identified the site using a chemical biology approach and then corroborated the exact predicted binding site [20] by mutational studies [31]. Thus, while the Arbidol binding site on S2 has been determined experimentally [31], the binding site of Clofazimine on the S2 segment remains unknown to the best of our knowledge. Along the lines of our previous efforts to predict small-molecule binding sites on S2 [19,20], we identify the most thermodynamically favorable binding site for Clofazimine using reliable CHARMM-based molecular docking methods. Using a recently published series of Clofazimine derivatives [23], we model the structure–activity relationship (SAR) and demonstrate that the data for 18 derivative compounds are better modeled at the proposed Site 2 rather than Site 1, which is the Arbidol binding site (Figure 3). We demonstrate experimentally that Arbidol and Clofazimine bind to the S2 segment. Finally, we use molecular modeling to identify the most favorable binding site for Clofazimine and illustrate how SAR data are best modeled at this proposed site.

## 2. Materials and Methods

Samples of Arbidol **1** and derivatives were purchased from ChemDiv, including **1** (1635-0087), **1b** (8015-5742) and **1c** (H027-0218C) and **1d** (H027-0205C). Other compounds were purchased from Selleck Chemicals, including Clofazimine **2** (S4107), Toremifene **3** (S1776), Ecliptasaponin A **4** (S9403) and Ivermectin (S1351). While Toremifene Citrate (S1776) and Ivermectin (S1351) came as a stock solution in 10 mM DMSO, all other purchased compounds were prepared as a standard 10 mM DMSO stock solution from an exact known weight (mg) of each compound. Compound DMSO stock solutions were prepared from the same source of DMSO used in the SPR experiments to minimize observed bulk responses in compound injections.

All SPR experiments were performed at Reaction Biology Corporation (Malvern, PA, USA) using a Biacore 8K+ (Cytiva) instrument with high sensitivity [32]. For immobilizations, a “series S” and “SA” sensor chips (Cytiva) were used to capture Avi-tag biotinylated protein samples of the SARS-CoV-2 Spike protein. Two biotinylated recombinant samples of the Spike protein were purchased from Acro Biosystems: a full-length trimeric SARS-CoV-2 Spike construct with the D614G mutation (Biotinylated SARS-CoV-2 S protein, catalog number SPN-C82E3) and a trimeric construct of the SARS-CoV-2 S2 segment (Biotinylated SARS-CoV-2 S2 protein, catalog number S2N-C52E8). The experimental design aims to compare a full-length Spike trimer to an equivalent trimeric S2 construct, and the two constructs utilized were the best available equivalents that were biotinylated and resulted in successful immobilizations and subsequent SPR binding experiments. Both constructs incorporate a series of substitutions that stabilize the folded trimeric prefusion conformation. The full-length construct with the D614G mutation (catalog number SPN-C82E3) contains Proline substitutions (F817P, A892P, A899P, A942P, K986P, V987P) to stabilize the trimeric prefusion state and Alanine substitutions (R683A and R685A) to remove the furin cleavage site. The S2 construct (catalog number S2N-C52E8) contains the same series of Proline substitutions (F817P, A892P, A899P, A942P, K986P, V987P) to stabilize the folded trimeric conformation and minimize the formation of misfolded aggregates during protein production. The S2 construct is shown by the vendor to reliably bind a Spike S2 subunit antibody (Human IgG1) by Avitag ELISA assays (Acro Biosystems, 1 Innovation Way, Newark, DE 19711, USA), where the full-length construct (catalog number SPN-C82E3) is shown by the vendor to reliably bind ACE2 and an Anti-Spike RBD neutralizing antibody (Human IgG1) (Cat. No. SAD-S35) by Avitag ELISA. The series of substitutions in both constructs act to stabilize the full-length folded trimer, as well as to stabilize the folded S2 trimer that reliably binds to a Spike S2 subunit antibody. Under our best conditions after immobilization, both constructs were able to demonstrate binding of small molecules, which diminished over time (after two hours) as is often observed for some folded proteins that may become unfolded over time once immobilized on the biosensor surface. Thus, while we do not know for sure if the exact folded structure of the two constructs is the same, SPR evidence from small-molecule binding experiments suggest that the immobilized samples are folded trimers rather than an unfolded immobilized peptide, which may aggregate on a biosensor surface.

Numerous attempts were made to prepare the biosensor surface and optimize the assay conditions to improve the quality of the binding data (See Section 3). The protein samples were immobilized on the SA chips using a 5 (mL/min) flow rate with a running buffer composed of PBS with 0.05% Tween 20, resulting in ranges of 3000–5000 RU. Following immobilization, both the sample surface and the reference channel surface were blocked with biotin to attempt to minimize non-specific binding. In the first round of assay development and initial data collection on the surfaces described above, small-molecule analytes were analyzed using running buffer composed of PBS with 0.05% Tween 20 and either a 1% DMSO or 2% DMSO solutions. Titrations of each analyte were performed using multi-cycle kinetics mode, with 200 mM as the highest concentration for a 2-fold serial dilution of 10 concentrations. In this first round of data collection, serial dilutions were performed on a plate, where the 200 mM concentration was prepared by mixing 10 mM DMSO stock solution with 0% DMSO running buffer to achieve either a 1% or 2% DMSO solution of analyte to match the running buffer.

In a final round of compound characterization using optimized conditions, a new SA chip was prepared, with the aim of facilitating collection of duplicate sensorgrams from two surfaces. Given the high lipophilicity of some of the compounds, rather than performing the dilution on the plate, the serial dilution was performed in DMSO first to avoid solubility issues. Ten concentrations were prepared in DMSO at 50× the concentration used in the assay and then transferred to the plate and mixed with the DMSO-free running buffer to achieve a 2% DMSO solution of analyte to match the 2% DMSO running buffer. Independent duplicates were compared for each compound and two separate concentration series were performed, with starting concentrations of 100 mM and 50 mM, respectively. Titrations of each analyte were performed using multi-cycle kinetics mode. All SPR data were appropriately solvent-corrected [33], reference-subtracted and analyzed while fitted to a steady-state affinity model using Biacore Insight Evaluation Software.

All molecular docking and free-energy calculations were performed using CHARMM [34] and the previously described CHARMM-based computational methods established by our laboratory [35,36]. Molecular docking utilized the LPDB CHARMm force field to model small-molecule potential functions and the resulting protein–ligand interactions [37,38]. As previously described, a two-step scoring approach was utilized to rank the final TOP5 poses from any docking attempt. For the final TOP5 docking poses, a final energy minimization of the protein–ligand complex was performed using the Generalized Born using Molecular Volume (GBMV) implicit solvent method [39,40]. Starting from the minimized complex, minimizations of the bound and free state were performed where potential energy components (VDW), (ELEC) and solvation (SOLV) were calculated in order to approximate the free energy of binding (∆G_bind_) using a linear interaction energy scoring approach with previously determined empirical generalized parameters [35]. Results using the predicted (∆G_bind_) values for the TOP5 poses of each individual docking “trials” were pooled and sorted by (∆G_bind_), and the top-ranked members of a geometric cluster (RMSD < 2.0 Å) were identified. Statistics for (∆G_bind_) were calculated from the average and standard deviation from the three top-ranked members of a geometric cluster (RMSD < 2.0 Å) or a triplicate representing the geometric cluster. For all work performed in this study, independent docking “trials” were initiated from 20 generated conformations of a given small-molecule ligand, such that the initial geometry was entirely independent of any CHARMM-based procedure. MarvinSketch version 15.8.31 is a publicly available 3D conformation generator that was used to generate non-identical low-energy conformations [41].

Our laboratory had previously used a pharmacophore procedure to identify the most favorable TOP50 binding sites on the SARS-CoV-2 Spike protein S2 segment [19]. This structural analysis was performed using the 3.2 Å CryoEM structure (6vyb.pdb) of the full-length Spike protein where the ectodomain was in the “closed” state (6vxx.pdb) [42]. From this model (6vxx.pdb), molecular docking was performed using a hierarchical approach, such that 10 conformations of Clofazimine were initially docked to all 50 sites on the Spike S2 segment. Then, after identification of the TOP5 most favorable sites from this first step, more extensive sampling was used to refine the ranking of the TOP5 sites, and 20 conformations of Clofazimine were docked. Using this model (6vxx.pdb), the consensus binding mode for Cofazimine binding to the SARS-CoV-2 Spike S2 segment (Figure 3B) was used to dock 18 derivatives of Clofazimine [23]. These derivatives were modeled at five binding sites: Site 1 and Site 2 on the S2 segment as shown in (Figure 3), as well as for the lowest-energy binding sites [19] in the Nsp5 Main Protease (6w63.pdb) [43], Nsp13 Helicase (6jyt.pdb) [44], and the Nsp16 2′-O methyltransferase (6wkq.pdb) [45]. The results for docking the series into the Nsp5 Main Protease and Nsp16 are expected to represent negative controls, where Clofazimine series SAR data would not be expected to show well-modeled binding to these two sites. As our previous work has highlighted, the Nsp5 Main Protease and Nsp16 binding sites in particular [19] are thermodynamically favorable for the binding of a variety of small-molecule fragment ligands. This makes them more challenging negative control “decoy” binding sites, particularly compared to most of the possible binding sites on the S2 segment, which are less thermodynamically favorable “decoy” binding sites. The physical basis for this is that the specific molecular shape and the hydrophobicity of the Nsp5 and Nsp16 binding sites are favorable for binding hydrophobic small molecules and result in more thermodynamically favorable and more negative (∆G_bind_) values when performing virtual screening of a library of compounds. All molecular graphics images of protein structures and molecular interactions were generated with UCSF Chimera [46].

## 3. Results

### 3.1. Biosensor Chip Preparation

Numerous attempts were made to prepare the biosensor surface and optimize the assay conditions to improve the quality of the binding data. The protein samples were immobilized on SA chips using a 5 (mL/min) flow rate with a running buffer composed of PBS with 0.05% Tween 20. For the full-length Spike protein on two separate channels, an injection of 20 (mg/mL) protein and a 200 s contact time resulted in an immobilization level of 2831.6 RU, where an injection of 40 (mg/mL) protein and a 600 s contact time resulted in an immobilization level of 5102.0 RU. For the S2 protein on two separate channels, an injection of 10 (mg/mL) protein and a 200 s contact time resulted in an immobilization level of 2831.6 RU, whereas an injection of 20 (mg/mL) protein and a 600 s contact time resulted in an immobilization level of 4721.0 RU. Even though both the sample surface and the reference surface were blocked with biotin to attempt to minimize non-specific binding, some non-specific binding was observed. While this non-specific analyte binding to the reference surface precluded characterization of some compounds, this effect was compound-specific and could be rationalized to specific compounds that are more hydrophilic and may contain more hydrogen-bonding opportunities in relation to the referenced biotinylated surface. In comparison, one of the most lipophilic compounds, Clofazimine (cLogP = 7.1), exhibited the least non-specific binding to the reference surface and reproducibly provided the highest-quality SPR binding data over all conditions explored. As expected, both the SPR binding signal (RU) and the overall quality of the SPR sensorgrams were best within the first 30 min to two hours after immobilizing both samples, as is commonly observed for folded proteins in SPR experiments.

In a final round of compound characterization using optimized conditions, a new SA chip was prepared with the aim of facilitating collection of duplicate sensorgrams from two duplicate surfaces. The protein samples were immobilized on the SA chips using a 5 (mL/min) flow rate with a running buffer composed of PBS with 0.05% Tween 20. For the full-length Spike protein, an injection of 40 (mg/mL) protein and a 600 s contact time resulted in duplicate immobilization levels of 4692.0 and 4754.6 RU. For the S2 segment, an injection of 20 (mg/mL) protein and a 600 s contact time resulted in duplicate immobilization levels of 3640.3 and 3758.0 RU.

### 3.2. Surface Plasmon Resonance (SPR) Binding Data

In preliminary rounds of assay development and initial data collection on the surfaces described above, small-molecule analytes were analyzed using running buffer composed of PBS with 0.05% Tween 20 and either a 1% or 2% DMSO solution. Titrations of each analyte were performed with 200 μM as the highest concentration for a 2-fold serial dilution of 10 concentrations. The best representative data for compounds **1**, **1b**, **1d** and **2** that fit to the steady-state affinity model are shown in Table 1. In the same SPR dataset as shown in Table 1, another negative control, small-molecule Ivermectin (MK-933), did not show any binding to either full-length Spike or S2, as expected. Some of the best representative data showing simultaneous SPR sensorgrams binding to Spike and S2 with good-quality fits are shown in Figure 4. For SPR data presented in Figure 4, all compounds demonstrate definitive sensorgram evidence for binding from concentration-dependent change in response units (RU) and good-quality fits to the steady-state affinity models. Figure 4 demonstrates compounds of four different structural classes that bind to the S2 segment as well as the full-length Spike protein. While some of the lower-affinity compounds, **1c**, **1d** and **4**, exhibited greater observed differences in affinities between S2 and Spike, the higher-affinity reference compounds **1**, **2** and **3**, all had reproducibly similar SPR sensorgrams and fits with lower differences in affinity between S2 and Spike, so for **1** (S2 = 5.9 μM) (Spike = 7.4 μM), for **2** (S2 = 6.3 μM) (Spike = 5.4 μM), and for **3** (S2 = 4.1 μM) (Spike = 4.1 μM). These binding data strongly suggest that the binding site of the small molecules is found on the S2 segment of the full-length protein.

The observed affinities in this round (K_d_ = 5.9 μM to 7.4 μM) for Arbidol **1** were well within the range of reported antiviral activities for Arbidol (EC50s = 4.1 μM to 10.0 μM) reported in the literature^1,2^ and in our previous work (EC50 = 5.6 μM) [20]. For **1**, which is a 6-Br derivative, as well as 6-(pyridin-2-yl) derivatives, **1c** and **1d**, it was generally observed that the affinity for binding to the S2 segment was slightly more favorable than that for binding to the full-length Spike protein, as observed in each individual compound. This is shown in Figure 2C, where **1** binds with a slightly higher affinity to the S2 segment (K_d_ = 5.9 μM) compared to the full-length Spike (K_d_ = 7.4 μM) shown in Figure 2B. The same trend (K_d_ S2 < K_d_ Spike) can be shown for 6-(pyridin-2-yl) derivatives **1c** and **1d** in Figure 4A,B, respectively. While the observed trend of (K_d_ S2 < K_d_ Spike) was not necessarily expected, it may be possible to rationalize the observation if the cleaved S2 segment small-molecule binding sites may be more dynamic or amenable to complementary induced-fit binding compared to the much larger full-length trimer.

An important observation was that when comparing the SPR data for either the full-length Spike or the S2 segment, the binding data exhibited the expected structure–activity relationship (SAR) for the derivative series such that for Spike, (**1** < **1c**) and (**1** < **1d**) [20]. Thus, **1** had a higher affinity than either 6-(pyridin-2-yl) derivatives **1c** or **1d,** as expected from both virtual screening data and experimental CPE data for **1** (EC50 = 5.6 μM) and the 4-chlorobenzenesulfonyl derivative **1b** (EC50 = 29.5 μM) [20]. The 4-chlorobenzenesulfonyl derivative **1b** combines two chemical substitutions that reduce activity and introduce less favorable and complementary protein–ligand interactions to the binding model, the 4-chloro substitution and the oxidation of the sulfanyl to the sulfone [20]. In addition, for only the data on the S2 segment, the same was observed: **1** had a higher affinity than either 6-(pyridin-2-yl) derivative **1c** or **1d,** as expected, providing additional confidence in the interpretation of the SPR data observed for different compounds. The observed affinity (K_d_ = 4.1 μM) for Toremifene **3** was very close to the reported antiviral activity for Toremifene in live SARS-CoV-2 infections (EC50 = 3.58 μM) [47] and (EC50 = 1.92 μM) for SARS-CoV-2 pseudovirus entry assays [48]. Observing the expected SAR for (**1**, **1c** and **1d**) and being in reasonable agreement with reported antiviral activity for **1** and **3** helps to establish that we are able to interpret the SPR results from more than one perspective.

In a final round of compound characterization, a new SA chip was prepared with the aim of facilitating collection of duplicate sensorgrams using optimized conditions and 2% DMSO running buffer. Independent duplicates were compared for each compound, and two separate concentration series were performed with a starting concentration of 100 μM and 50 μM, respectively. This dataset resulted in five compounds with quality duplicates binding to the S2 segment. The best SPR duplicates for binding to the S2 segment are shown in Figure 5. The best representative data for compounds **1**, **1c**, **2**, **3**, and **4** and fits to the steady state affinity model are summarized in Table 2; statistics are presented for duplicates. While the duplicate affinity of **1** binding to S2, as reported in Table 2, is slightly lower (K_d_ = 2.1 ± 0.2 μM) in this dataset than (K_d_ = 5.9 μM), as reported in Table 1, it remains much lower than the 6-(pyridin-2-yl) derivative **1c** (K_d_ = 11.4 ± 1.3 μM). Thus, the binding data in the duplicates also follow the expected SAR for the derivative series, such that for binding to S2, **1** had a higher affinity than derivative **1c,** as expected [20]. Again, in this dataset of duplicate measurements, the observed affinity for Spike (K_d_ = 4.1 mM) for Toremifene **3** was very close to several reported antiviral activities (EC50s = 1.9 μM to 3.6 μM) [47,48].

As mentioned previously, Clofazimine was found to provide the most reliable and highest-quality SPR binding data to S2 over the conditions explored. In a recently published series of Clofazimine derivatives, the affinity of **2** to full-length Spike by SPR was found to be (Spike = 3.82 μM) [23], which was very close to the best representative data for 2 (S2 = 3.9 μM) (Spike = 2.9 μM) from the first round, as presented in Table 1. The best duplicate data for **2** (S2 = 6.5 ± 0.3 μM) (Spike = 4.6 ± 1.2 μM) are presented in Table 2. While the affinity is a bit lower in the duplicate dataset, both datasets show that Clofazimine definitively binds to the S2 segment under these conditions.

While Clofazimine has been reported to have a range of potent antiviral activity in CPE assays ranging from (EC50 ~ 0.08 μM to 0.56 μM) [4,5,6,7] with a consensus of (EC50 = 0.31 μM) [4,5], the antiviral activity of **2** has been reported to result as a combination of Spike-dependent fusion inhibitor activity as well as Nsp13 helicase unwinding activity [5]. The dose–response curves of **2** in the same study [5] suggest that the micromolar viral fusion activity (EC50 ~ 2.5 μM to 5.0 μM) is slightly more potent than the Nsp16 helicase unwinding activity (EC50 ~ 7.5 μM to 10 μM) [5].

Interestingly, in other independent assays that report SARS-CoV-2 Spike-mediated fusion activity [49], the activity values also show lower potency for Clofazimine (EC50 = 2.56 μM) [49], which is much closer to the current measured affinity for full-length Spike (K_d_ = 2.9 μM to 4.6 μM) by SPR and another reported value (K_d_ = 3.82 μM) for full-length Spike by SPR [23]. In summary, while the observed affinity by SPR binding assay for Clofazimine is in the micromolar range (K_d_ = 2.9 μM to 4.6 μM) rather than the more potent observed antiviral activity (EC50 = 0.31 μM) [4,5], this is in agreement with observations from viral fusion assays [5,49], SPR binding [23] and the concept that the resulting antiviral activity is a result of dual-targeted drug action on at least Spike and the Nsp13 helicase [5]. While Clofazimine has been reported to be a viral fusion inhibitor, to our knowledge it has yet to be reported that Clofazimine binds to the S2 segment of Spike. As Clofazimine is an important clinical candidate, narrowing down its mode of action as a direct-acting fusion inhibitor is important. The SPR data show that Clofazimine 2 binds to a well-formed binding site on the S2 segment trimer.

### 3.3. Predicting the Clofazimine Binding Site on S2 with Molecular Docking

While the Arbidol binding site has been experimentally determined by Shuster et al. [31], there has yet to be any published experimental structure of a small-molecule fusion inhibitor bound to the Spike S2 segment solved by either X-ray crystallography or CryoEM techniques. From docking **2** into all the TOP50 binding sites predicted on the S2 segment [19], the top two favorable sites were identified and shown in Figure 6A. Site 2, shown in (Figure 3B), is the only feasible binding site for **2** according to our modeling data (Figure 6B), whereas Site 1 is predicted to be much less thermodynamically favorable for binding of 2. From analysis of molecular docking and calculated (∆G_bind_) values at all 50 sites [19,20], Site 2 is easily identified as being the most favorable site, based on the identification of two other structurally related 3-fold symmetric sites. In terms of predicted (∆G_bind_) values from the statistics of the top-ranked cluster (as a triplicate), Site 2 (∆G_bind_ = −9.4 ± 0.3 kcal/mol) is much more thermodynamically favorable than Site 1 (∆G_bind_ = −7.6 ± 0.2 kcal/mol), the Arbidol binding site. The protein–ligand interactions of Clofazimine, modeled at Site 2, are complementary and favorable, as described in more detail in the next Section 3.4. In summary, as shown from the docking and predicted (∆G_bind_) values (Figure 6B), Clofazimine and **3** are predicted to bind more favorably at Site 2 compared to Site 1. Ecliptasaponin A **4** is predicted to bind favorably at Site 1, the Arbidol binding site (Figure 6A). We have previously demonstrated how a series of oleanolic acid (OA) Saponin derivatives are best modeled at Site 1 rather than Site 2 on the S2 segment [20], and Ecliptasaponin A is closely related in structure to OA Saponin derivatives such as **12a**. Both **4** and **12a** are predicted to bind more favorably at Site 1, the Arbidol binding site, as shown in Figure 6.

### 3.4. Modeling a Series of Clofazimine Derivatives Binding to the S2 Segment

Beyond the fact that Clofazimine is predicted to bind much more favorably to Site 2 than Site 1, according to calculated (∆G_bind_) values, another independent line of evidence from modeling also strongly corroborates Site 2. Recently, a new series of chemical derivatives of Clofazimine were published with antiviral activity data against SARS-CoV-2 [23]. Using the same methods, 18 derivatives were modeled binding to Site 1 and Site 2. For each derivative in the series, a TOP-ranked cluster was determined independently from docking numerous initial starting conformations, rather than simply modeling all derivatives exactly as the binding mode of the reference compound.

In modeling the series of 18 derivatives at Site 2, the “untrained” predicted ∆G_bind_ values exhibited some correlation with the experimental EC50 values. The Pearson’s R^2^ correlation coefficient was R^2^ = 0.264 for all 18 compounds (Figure 7A) modeled at Site 2. In comparison, 18 compounds (Figure 7A) modeled at Site 1 exhibited a positive correlation but with a very low calculated correlation coefficient R^2^ = 0.029. Thus, from modeling all 18 compounds, the “untrained” predicted ∆G_bind_ values had much greater correlation at Site 2 (R^2^ = 0.264) compared to Site 1 (R^2^ = 0.029). Compared to previous benchmark studies characterizing this scoring function method and performance against datasets of diverse protein binding site architectures and protein–ligand interactions, these levels of R^2^ correlation are adequate to establish confidence in the binding model as reflecting experimental SAR data [20,35,36] compared to models with zero correlation (R^2^ = 0.0). While the robustness of this correlation analysis may be determined rigorously using a cross-validation approach [20], this is not required in this situation, as the series may also be easily modeled as two separate series of derivatives. One series is structurally related to reference compound Clofazimine (**6d**, **6e**, **7a**, **7b**, **7c**, **7d**, **7e**, **7f**, **7g**, **7i**, **7k**, **7m**, **7o**) and the other series is based on a different reference compound substructure (**15a**, **15b**, **15f**, **15b**, **15h**). Pearson’s R^2^ correlation values ranged from R^2^ = 0.247 for all 18 compounds (Figure 7A) modeled at Site 2, where even higher correlation coefficients of R^2^ = 0.306 to 0.311 were achieved modeling the dataset as these two separate series of “untrained” predicted ∆G_bind_ rankings as shown in (Figure 7B) with similar slopes. Compared to previous studies using this approach, the observed R^2^ correlation and slope for the two series are sufficiently similar [20,35,36].

As shown in more detail in Figure 8, the derivatives from both series are well modeled at Site 2 and the binding mode can rationalize the SAR functional group substitutions at all three R groups (**R**, **R_1_** and **R_2_**). The model can rationalize the SAR relationship at **R,** where (O-CH_3_ > Cl > F). The reference Cl atom forms not hydrophobic interactions, but rather close and favorable hydrophilic interactions with the positively charged NZ atom from the side chain of K1038, where the phenyl ring forms favorable hydrophobic interactions with the hydrophobic side chain of K1038 atoms (CB, CG, CD, CE). Thus, the **R** group is partially solvent-exposed in close proximity to electrostatic interactions with the NZ atom side-chain of K1038. The substitution O-CH_3_ forms favorable interactions, but the F atom exhibits a weaker molecular interaction with a positively charged NZ atom than Cl. Thus, the model is able to rationalize the most important substitutions leading to favorable **R** groups.

Next, the model can explain the series of substitutions at **R_1_**, where the phenyl ring is buried in a hydrophobic pocket formed primarily by the side chain of A890 and Y1047 on one side and V1040 on the other side. For the position of the **R_1_**, para Cl or F substitutions are found in more favorable derivatives such as **15g**. The favorability of F over Cl is easily rationalized by its proximity at the back of a hydrophobic pocket with close interactions with dipolar backbone atoms 2.54 Å from (G1046@HN) and 3.19 Å from (D141@OD1). Other **R_1_** substitutions are also rationalized in this binding mode, such as O-CF_3_ (**7i**) being more favorable than O-CH_3_ (**7g**), where one of the CF_3_ electronegative fluorine atoms of (**7i**) forms a favorable electrostatic interaction with a backbone amide H1048@HN, such that the isosteric CH_3_ substitution is less favorable. Finally, the model is also able to rationalize the series of substitutions at **R_2_**, namely that the isopropyl group is more favorable than the cyclopropyl, as demonstrated for the derivatives **7a** and **15h**. For derivatives **7a** and **15h**, the cyclopropyl group carbon atoms are more unfavorable as they are closer in distance to the polar side-chain atoms of R1107 and N1108. The smaller isopropyl group lacks these unfavorable interactions and the carbon atoms bind a bit closer in distance to the aromatic carbon atoms of W886. In summary, the derivative series when modeled at Site 2 forms complementary protein–ligand interactions that are able to explain substitutions at **R**, **R_1_** and **R_2_**.

### 3.5. Modeling a Series of Clofazimine Derivatives Binding to Other SARS-CoV-2 Targets

To increase confidence in our comparison with the derivative series SAR data, the series of derivatives were also independently docked at other binding sites of other SARS-CoV-2 target proteins. As mentioned previously, Clofazimine has been reported to be a dual-targeted SARS-CoV-2 antiviral [5], with Spike-dependent fusion inhibition activity as well as Nsp13 helicase unwinding antiviral activity [5]. Interestingly, the same research group also measured zero activity for Clofazimine in an assay for the Nsp5 Main protease (Mpro) activity [5]. As we had previously published maps of thermodynamically favorable binding sites for Nsp5 Mpro, Nsp13 helicase and Nsp16 2′-O methyltransferase [19], we selected to model the derivative series at the most favorable site identified on these targets for Clofazimine. Thus, Nsp5 Mpro and Nsp16 are “negative control” proteins, where we would expect no correlation with experimental SAR data, particularly since **2** has been reported to have no inhibition activity for Nsp5 Mpro. As expected, modeling the series of 18 derivatives at both Nsp5 Mpro and Nsp16 as “negative control” binding sites resulted in poor agreement with the experimental SAR data, as well as less favorable predicted (∆G_bind_) values. Modeling the series at Nsp5 Mpro, the “untrained” predicted ∆G_bind_ values exhibited a negative correlation (a negative slope) with a very low correlation coefficient R^2^ = 0.014 for all 18 compounds (Figure 7A). This agrees with the observation that Clofazimine has been reported to have no inhibition activity for Nsp5 Mpro [5]. Modeling the series at Nsp16, the “untrained” predicted ∆G_bind_ values exhibited a negative correlation with a low correlation coefficient R^2^ = 0.056 for all 18 compounds (Figure 7A). Interestingly, the results in Figure 7A show that in modeling the series of 18 derivatives at the most favorable site identified for Clofazimine on the Nsp13 helicase (see Appendix A), the “untrained” predicted ∆G_bind_ values did exhibit some correlation (R^2^ = 0.141) with the experimental EC50 values, but not as much correlation as Site 2 on the S2 segment (R^2^ = 0.264).

To summarize, the comparison of the docking data at other target proteins “decoy” binding sites also strengthens the conclusion that the series of Clofazimine derivatives are best modeled at Site 2 on the S2 segment, rather than Site 1 on the S2 segment. When the series is modeled at all 5 binding sites, the only sites that have reasonable R^2^ correlation values and positive slopes are for binding at Site 2 on the S2 segment (R^2^ = 0.264) and at the Nsp13 helicase site (R^2^ = 0.141).

## 4. Discussion

### 4.1. Possible Implications for Broad-Spectrum Antiviral Activity

As the COVID-19 pandemic progressed, it was not surprising that most of the observed mutations to the SARS-CoV-2 Spike protein were found in the S1 segment of Spike, which contains the receptor binding domain (RBD). In comparison, fewer mutations have been found on the S2 segment, which exhibits greater conservation in sequence across coronavirus strains [50]. The result that Clofazimine binds to the S2 segment of Spike might have been anticipated from the sequence alone, as **2** has been shown to exhibit some broad-spectrum activity against other coronavirus strains such as MERS [5]. When the proposed Clofazimine binding site is superimposed with other experimentally determined coronavirus structures in structure–sequence alignments, the changes in sequence and structure are able to rationalize the broad-spectrum antiviral activity of **2** in closely related coronaviruses, including SARS CoV, MERS, hCoV-229E and hCoV-OC43.

Figure 9B shows the backbone superposition in a structure–sequence alignment of SARS-CoV-2 and MERS structures, where Figure 9D shows the complementary fit of the molecular surface. Figure 9E shows how the MERS binding site is still formed, with minimal atom-clashes with the model of bound Clofazimine. Figure 9F shows the sequence conservation in the residue segments that form the binding site. In the highly conserved SARS-CoV-2 sequence 1036–1040 (QSKRV), Q1036 is the most conserved residue that forms the Clofazimine binding site with the (i, i + 2) residue K1038, which is a key residue that forms important hydrophobic and hydrophilic interactions with **2** in the model. In following the sequence conservation of the position K1038, the residue is the same for the most closely related viruses (SARS-WH20, SARS-BJ01, and MERS) and then begins to diverge, with the mutation K1038S for Human coronavirus OC43 (7sb3.pdb) [51], and K1038P for human coronavirus HKU1 (8ohn.pdb) [52]. Interestingly, Cofazimine has been shown to have some activity [2,5] for the strain hCoV-229E (6u7h.pdb) [53], and this sequence retains the K1038 residue which is key to the binding site [53]. Clofazimine has been shown to have some activity for the strain human coronavirus OC43 [2,5], where the binding site is perturbed from the substitution K1038S and the binding site model would predict much lower activity for the OC43 strain compared to SARS-CoV-2. This trend has been observed experimentally [2], where **2** was found to be less potent in infections with hCoV-OC43 (EC50 = 0.35 μM) compared to SARS-CoV-2 (EC50 = 0.01 μM) in the same study [2]. As the sequence diverges further for the residues forming the binding site, the model would predict diminishing activity in other more distantly related coronaviruses such as Rhinolophus bat coronavirus HKU2 (6m15.pdb) [54]. Put simply, the proposed binding model seems to be able to account for current information from numerous experimentally determined structures of Spike and available antiviral activity for **2** against several coronavirus strains (SARS-CoV-2 compared to MERS or OC43).

While it is quite possible that several other research groups have independently identified this putative binding site on the S2 segment, to the best of our knowledge, this binding site was first highlighted in the literature by our studies [19,20] and was also recently independently identified by Zannella et al. [55], using an entirely different genetic approach to identify short peptide inhibitors of SARS-CoV-2 [55]. The short tripeptide inhibitor VFI was identified experimentally by Zannella et al. They proposed that VFI binds to the current site on S2 and demonstrated greater antiviral effects in pre-treatment assays; similar to a fusion inhibitor [55]. Interestingly, unlike the other peptides identified in that study, the VFI peptide exhibited broad-spectrum antiviral activity for both SARS-CoV-2 and hCoV-OC43 [55], similar to Clofazimine. When the VFI tripeptide is docked to Site 2, it binds in a very similar binding mode as Clofazimine, with impressive superposition of the three major hydrophobic peptide side-chain pharmacophores (Figure 9C).

### 4.2. Possible Implications for Spike-Dependent Mechanism of Action as a Fusion Inhibitor

For the proposed Clofazimine binding site (Site 2), a structural comparison of the experimentally determined structures of the prefusion and post-fusion conformation [56] provides a model for the direct action of Clofazimine on S2. Figure 10A depicts a model of Clofazimine bound to the experimental structure of the prefusion conformation and then superimposed on the experimentally determined post-fusion structure of S2 [56]. Figure 10B illustrates how the Clofazimine binding site undergoes a significant conformational change associated with the hydrophobic collapse of the binding pocket in the post-fusion conformation of S2. While Clofazimine has favorable and complementary protein–ligand interactions in the prefusion conformation, the modeled conformation of **2** in the post-fusion structure results in significant atom clashes, as shown in Figure 10B, from the resulting refolding and hydrophobic collapse of the local elements of protein structure. Clofazimine binding at this site is highly favorable in the prefusion conformation due to complementary hydrophobic interactions, which should result in a ligand-induced stabilization of the prefusion conformation of S2. The hydrophobic collapse of this site in the post-fusion conformation should prevent binding of **2,** according to our model. The model suggests that Clofazimine binding in the prefusion conformation is the most likely mode of action, stabilizing the prefusion state and preventing conformational changes within S2 that are required for membrane fusion.

## 5. Conclusions

Clofazimine has been shown to be a potent SARS-CoV-2 fusion inhibitor with robust activity in numerous antiviral assays [2,3,4,5,6,7,8,9,10]. While Arbidol was only found to be a partial inhibitor in SARS-CoV-2 in cytopathic effect (CPE) assays [20], in sharp contrast, several studies have shown that Clofazimine is a full inhibitor of SARS-CoV-2 infections in a range of cell types (Vero E6, Huh7, and Caco-2 cells) [4,5,6] and physiologically relevant cell lines (cardiomyocytes, Calu-3 and human primary airway epithelial cells) [5,8]. Clofazimine has also demonstrated promising preclinical antiviral activity in a Syrian hamster animal model of SARS-CoV-2 infection [5]. As Clofazimine has demonstrated synergistic antiviral activity with other direct-acting antivirals, such as Remdesivir [5], we hope that fusion inhibitors with this mechanism of action may be considered for future synergistic drug combination therapies [15]. More recent mechanistic studies have also shown that Clofazimine is able to inhibit Spike-induced activation of TMEM16 and subsequent procoagulant activity [57]. This observation may increase clinical interest in using Clofazimine as an experimental drug in the treatment of COVID-19 infections with significant pulmonary thrombosis, or in the treatment of a range of other Spike-induced pathologies, potentially including the treatment of “long COVID” [58,59,60,61,62,63,64].

While previous studies have extensively characterized the SARS-CoV-2 antiviral activity of Arbidol [1,2] and Clofazimine [2,3,4,5,6,7,8,9,10], to the best of our knowledge, no other group has yet reported biophysical binding data for Arbidol or Clofazimine binding to S2. While Clofazimine has been previously shown to bind to the SARS-CoV-2 Spike protein [23], it has yet to be reported that Clofazimine binds to the S2 segment of Spike. This experimental result is important for understanding the effects of fusion inhibitors of different structural classes and their specific mechanisms of inhibiting viral fusion. Aiming to avoid viral resistance mutations, the binding sites described on the S2 segment are composed of very conserved residues that seem to be required for S2 fusion activity function [50]. While it has yet to be experimentally determined exactly where Clofazimine binds to S2, we provide several lines of evidence that Clofazimine is best modeled as binding at Site 2. While it is only a model, it is a plausible structural hypothesis that is very useful to guide our next round of experimental design, including mutagenesis with complementary biophysical and pseudovirus entry assays. In light of the relatively low binding affinity (6.5 µM) of Clofazimine for S2 and the absence of evidence confirming the direct interaction between Clofazimine and the predicted binding site, it still remains premature to exclude other possible binding sites. As Clofazimine has been demonstrated to be one of the most promising clinical fusion inhibitors of Spike, we hope that this work will provide important structural insight for developing improved fusion inhibitors that will target S2 and elucidate the mechanism of direct drug action.

## Figures and Tables

**Figure 1 viruses-16-00640-f001:**
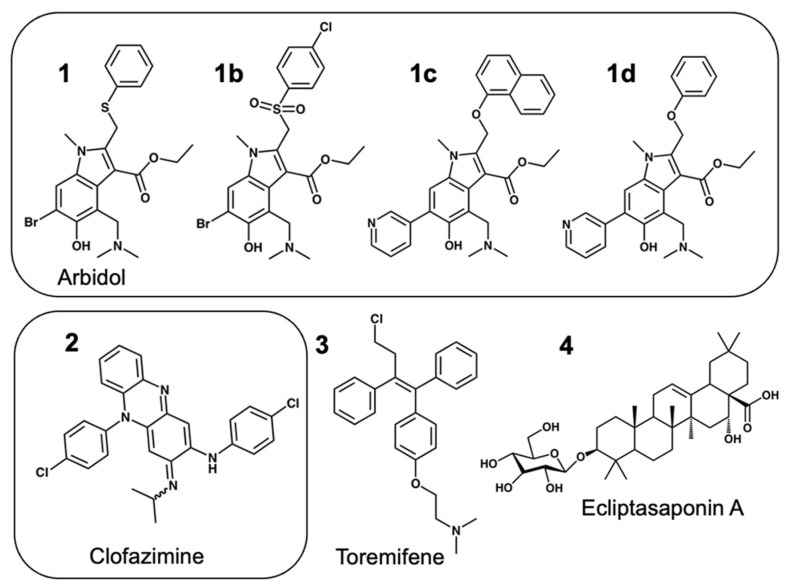
Structure of fusion inhibitor derivatives used in this study.

**Figure 2 viruses-16-00640-f002:**
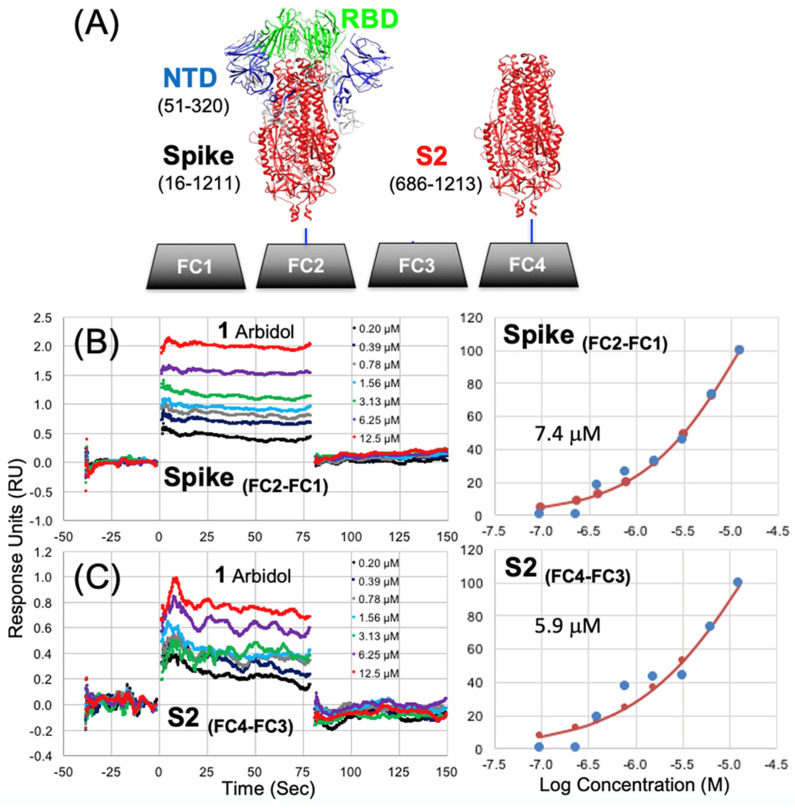
A direct SPR binding assay measures simultaneous binding to full-length Spike and the S2 segment. (**A**) Flow cell (FC) surfaces shown with attached Spike (FC2), reference FC1 and the S2 segment (FC4) and reference FC3. (**B**) Arbidol **1** binding to full-lennth Spike and (**C**) Arbidol **1** binding to the S2 segment showing both SPR sensorgrams and steady-state-affinity models.

**Figure 3 viruses-16-00640-f003:**
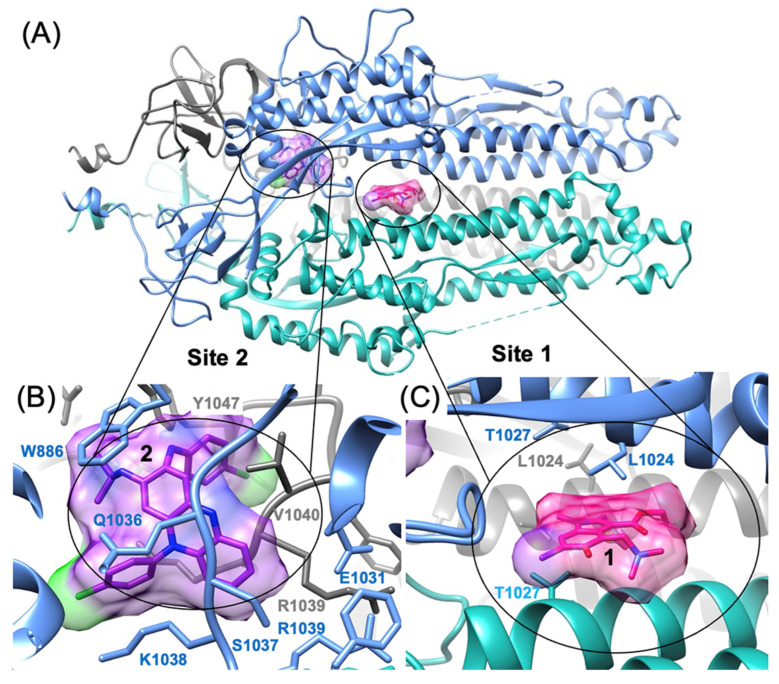
Structure of the SARS-CoV-2 Spike S2 segment showing two possible fusion inhibitor binding sites. (**A**) The trimeric S2 segment in a pre-fusion conformation showing (**B**) Site 2 with Clofazimine **2** bound and (**C**) Site 1 with Arbidol **1** bound.

**Figure 4 viruses-16-00640-f004:**
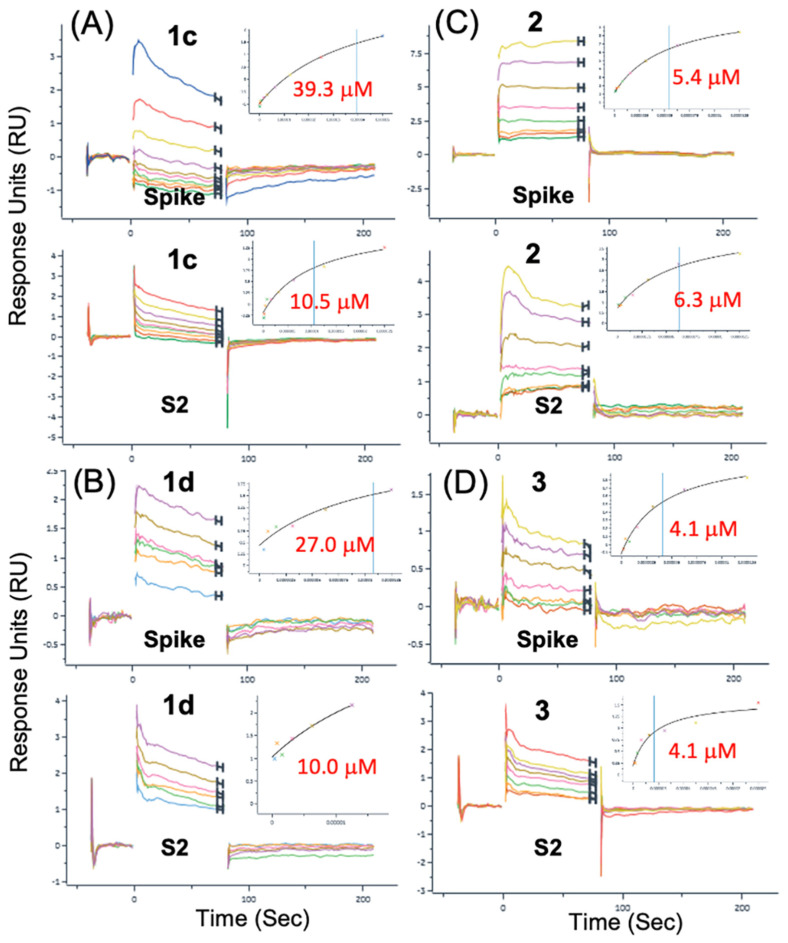
SPR data for fusion inhibitors binding to Spike and the S2 segment. Compounds from five different structural classes were shown to bind to the S2 segment. (**A**) Arbidol derivative **1c**, (**B**) Arbidol derivative **1d**, (**C**) Clofazimine **2**, (**D**) Toremifene **3**.

**Figure 5 viruses-16-00640-f005:**
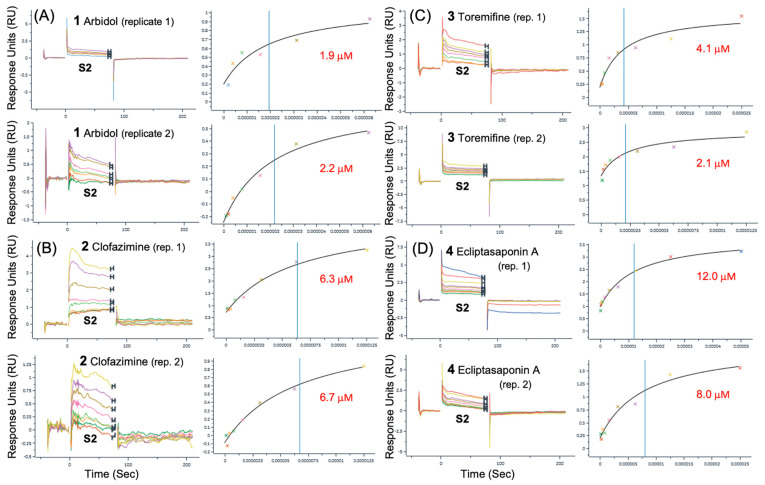
SPR duplicates for fusion inhibitors binding to the S2 segment. Duplicate binding curves for each compound are shown comparing only binding the S2 segment of Spike for (**A**) Arbidol **1**, (**B**) Clofazimine **2,** (**C**) Toremifene **3** and (**D**) Ecliptasaponin A **4**.

**Figure 6 viruses-16-00640-f006:**
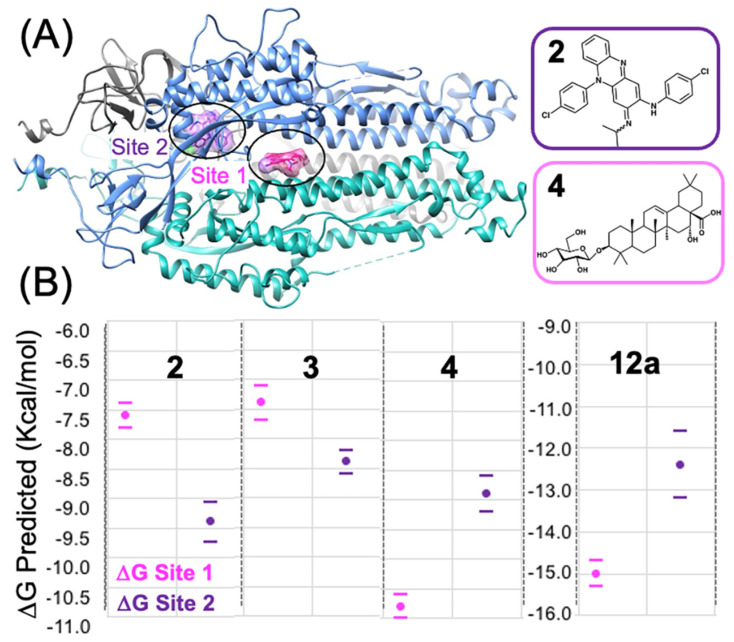
Modeling fusion inhibitors at two possible binding sites on the S2 Segment. (**A**) Shown is the trimeric S2 segment in a prefusion conformation showing Site 2 with bound Clofazimine highlighted in purple and Site 1 with bound Arbidol highlighted in magenta. (**B**) For each fusion inhibitor, ∆G_bind_ values are calculated based on the lowest-energy cluster modeled at Site 1 and Site 2, shown in magenta and purple, respectively. Clofazimine **2** and Toremifene **3** are predicted to bind more favorably to Site 2, while Ecliptasaponin A **4**, and OA Saponin **12a** are predicted to bind more favorably to Site 1.

**Figure 7 viruses-16-00640-f007:**
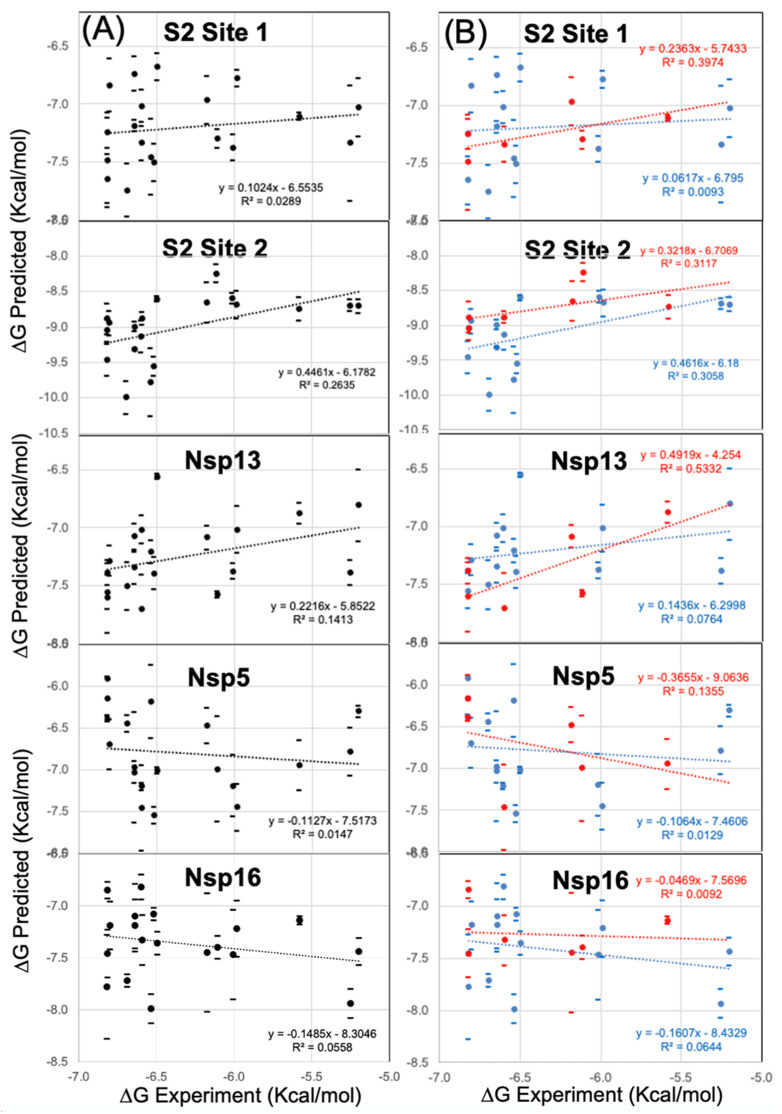
Predicted ∆G_bind_ values from Site 2 exhibit correlation with experimental SAR data for a series of Clofazimine derivatives. A series of 18 Clofazimine derivatives were well modeled as binding to Site 2, where sufficient linear correlation was achieved either comparing (**A**) all 18 derivatives (R^2^ = 0.264) or (**B**) two separate groups of compound series (R^2^ = 0.311) and (R^2^ = 0.306) with experimental SAR data. Poor correlation was observed when the compounds are modeled at Site 1 of S2, Nsp5, or Nsp16 “decoy” binding sites. When the series was modeled at the most favorable site on the Nsp13 helicase, the predicted ∆G_bind_ values exhibited some level of correlation for all 18 derivatives (R^2^ = 0.141) and quite reasonable correlation for the series of (**15a**, **15b**, **15f**, **15b**, **15h**) (R^2^ = 0.533).

**Figure 8 viruses-16-00640-f008:**
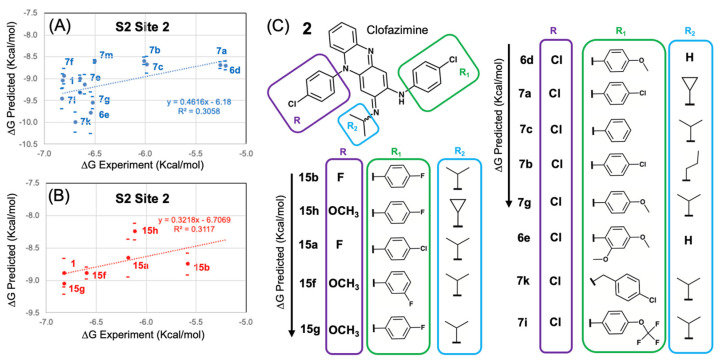
Predicted ∆G_bind_ values from Site 2 correlated with experimental SAR data and explained SAR substitutions for all three R groups. A series of 18 Clofazimine derivatives were well modeled binding to Site 2 as two separate groups of compound series where the first series (**A**) shown in blue (**6d**, **6e**, **7a**, **7b**, **7c**, **7d**, **7e**, **7f**, **7g**, **7i**, **7k**, **7m**, **7o**) with correlation (R^2^ = 0.306) and the second series (**B**) shown in red (**15a**, **15b**, **15f**, **15b**, **15h**) had a slightly greater correlation (R^2^ = 0.311) with experimental SAR data. (**C**) A diagram to illustrate how the series modeled at Site 2 rationalized **R** group substitutions at **R** (purple), **R_1_** (green) and **R_2_** (cyan).

**Figure 9 viruses-16-00640-f009:**
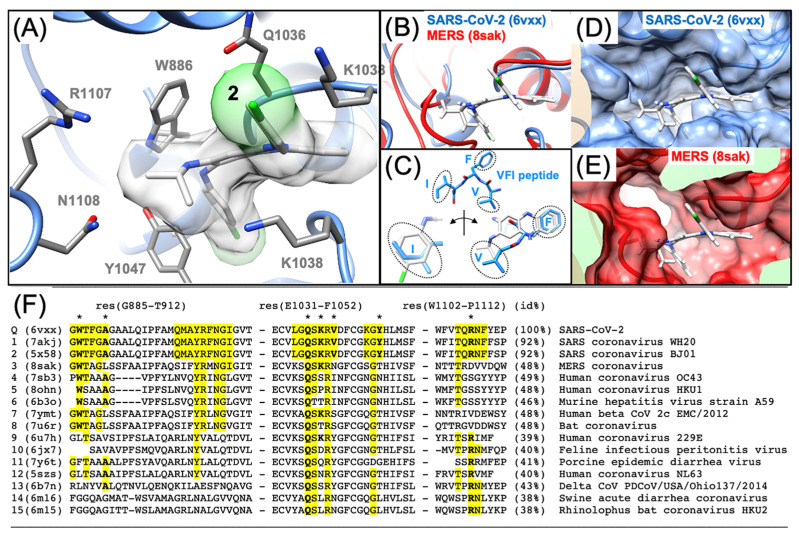
The predicted Site 2 is consistent with available structural information and may rationalize broad-spectrum activity of Clofazimine for MERS and other coronaviruses. (**A**) Predicted binding site for Clofazimine is shown illustrating the key binding site residues (W886, Q1036, K1038, V1040, and Y1047) shown below in the structure–sequence alignment. (**B**) A ribbon diagram is shown from a structure–sequence alignment between SARS-CoV-2 (6vxx.pdb) and MERS (8sak.pdb). (**C**) When docked at Site 2, the VFI tripeptide shows pharmacophore overlap with Clofazimine and the three major hydrophobic peptide side-chain pharmacophores. The surface model of the binding site is shown in (**D**) for SARS-CoV-2, showing a highly complementary binding surface for bound Clofazimine 2 in blue, where (**E**) shows that the binding surface shown in red is very similar in MERS with few atom clashes with the Clofazimine 2 binding mode. (**F**) Shows a sequence alignment derived from structure–sequence alignments with experimentally determined structures of the Spike protein from 15 different coronavirus strains. The sequence conservation of the SARS-CoV-2 residues that form the binding site (W886, A890, Q1036, K1038, V1040, Y1047 and R1107) are highlighted and denoted with the * symbol, where Q1036 is the most conserved of these residues.

**Figure 10 viruses-16-00640-f010:**
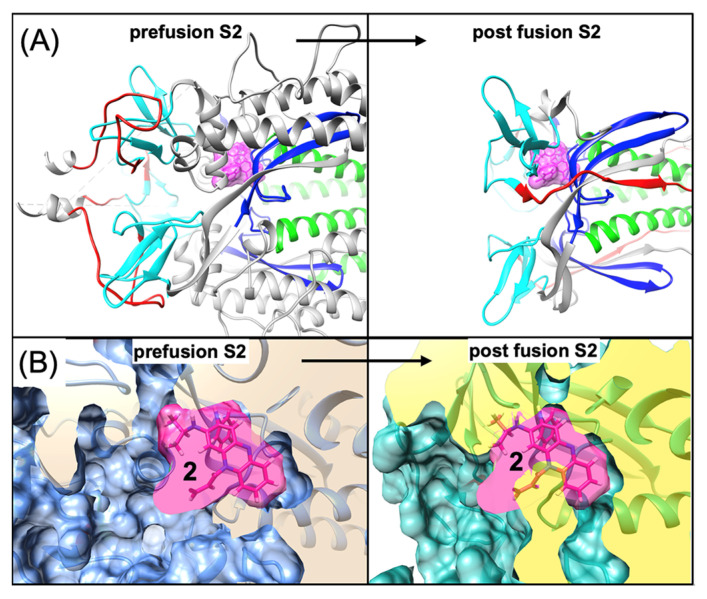
A possible mechanism of action for binding at Site 2 is to stabilize the prefusion conformation and prevent conformational changes required for fusion. (**A**) Ribbon diagrams of the experimentally determined structures of the prefusion conformation and the superimposed post-fusion structure (6xra.pdb) of S2. The proposed binding site for Clofazimine **2** is highlighted with a magenta molecular surface. To visualize local conformational changes, four residue segments (943–1034), (1035–1070), (1078–1120) and (1121–1141) are shown as green, blue, cyan and red, respectively. (**B**) A zoomed-in molecular surface diagram showing the superimposed structure of the prefusion conformation in medium blue showing complementary molecular surface and interactions where, in the post-fusion structure, the magenta atoms and surface of **2** clash with the teal and yellow molecular surface that has undergone local hydrophobic collapse during the conformational transition.

**Table 1 viruses-16-00640-t001:** Preliminary SPR binding data for binding to Spike and to the S2 segment. The best representative binding data from round 1 are shown.

	Spike	Spike	S2	S2
cmp	K_d_ (μM)	Affinity Chi^2^ (RU^2^)	K_d_ (μM)	Affinity Chi^2^ (RU^2^)
**1**	7.44	1.04 × 10^−2^	5.9	4.1 × 10^−3^
**1b**	N/A	N/A	31.2	4.06 × 10^−3^
**1d**	10	2.47 × 10^−2^	27	2.35 × 10^−2^
**2**	2.9	9.46 × 10^−3^	3.9	4.93 × 10^−2^

**Table 2 viruses-16-00640-t002:** SPR binding data duplicates for binding to the S2 segment. The best representative binding data from round 2 are shown with the calculated standard deviation for duplicates.

	Spike	Spike	S2	S2
cmp	K_d_ (μM)	Affinity Chi^2^ (RU^2^)	K_d_ (μM)	Affinity Chi^2^ (RU^2^)
**1**	N/A	N/A	2.1 ± 0.2	1.43 × 10^−3^
**1c**	40.4 ± 1.5	6.10 × 10^−3^	11.4 ± 1.3	9.86 × 10^−3^
**2**	4.6 ± 1.2	5.34 × 10^−3^	6.5 ± 0.3	1.30 × 10^−2^
**3**	4.1	2.70 × 10^−3^	3.1 ± 1.4	1.69 × 10^−2^
**4**	73.8 ± 8.3	2.69 × 10^−3^	10.0 ± 2.8	1.04 × 10^−2^

## Data Availability

The program CHARMM is publicly available under academic license for research (https://www.charmm.org, accessed on 5 January 2021). MarvinSketch is publicly available under academic license for research (http://www.chemaxon.com). UCSF Chimera is publicly available under academic license for research (https://www.cgl.ucsf.edu/chimera). All relevant data are shown in figures, listed in tables or included in the Appendix A. The docking ∆G_bind_ data and 2D compound information for all compounds may be found in the manuscript and in Appendix A. PDB files output files are provided in the Appendix A as a .ZIP file.

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
