# Peer review of "The Dual-Targeted Fusion Inhibitor Clofazimine Binds to the S2 Segment of the SARS-CoV-2 Spike Protein"

_viruses, 2024, doi:10.3390/v16040640_

Round 1
Reviewer 1 Report
Comments and Suggestions for Authors
Clofazimine and Arbidol have both been reported to be effective in vitro SARS-CoV-2 fusion inhibitors in previous report. In that report, Clofazimine and Arbidol showed binding of S2 segment of Spike protein?
In there any viral infection, Clofazimine and Arbidol had antiviral effect or not? If it has, please add discussio part.
How about antiviral effect of Clofazimine and Arbidol inSARS-CoV, MERS, hCoV-229E, hCoV-OC43 in vitro or vivo?
Based on this result, Clofazimine and Arbidol had antiviral effect targeting on Spike protein? If so, is there any measurment of antiviral effect Clofazimine and Arbidol on Spike protein?
How do you think that Clofazimine and Arbidol can use antiviral effect on treatment of COVID 19?
Reviewer 2 Report
Comments and Suggestions for Authors
ESR showed that Clofazimine 2 binds more predominantly to the spike protein than several competing reagents, and the binding sites were further estimated by calculation. Compared to the robustness of the ESR data, the computational part of the study is a little worrying, however, the findings will be useful to further improve the drug in the future.
One order of advice: clofazimine 2 has been in use for a long time and the side effects are well known. Please mention this, even as a warning to those who actually intend to use it. It colours the skin and it may not disappear for several years. However, this medicine is much cheaper to obtain than Lemdecivir or antibody-based medicines. This can be good news for many poor people. Please introduce this as well.
Reviewer 3 Report
Comments and Suggestions for Authors
Current work studies binding properties of some drugs being used in different countries for Covid-19 treatment to the S2 spike subunit of SARS-CoV-2.
The authors apply SPR to calculate binding efficiency, and docking to search for the binding pocket. The study could be of interest but should be further improved prior to possible publication
Major concerns
-
I am slightly concerned about the structural identity of the S2 subunit within the full-length Spike with D614G and as a separate peptide. How do you know that (lines 131-133) “As the D614G mutation is contained within S1, rather than in the S2 segment, the S2 construct would correspond exactly to the maturated S2 segment from either WT or the D614G full-length Spike protein”?
-
What negative and positive controls were taken for SPR?
-
Also, I failed to understand why in the beginning you show and discuss arbidol binding, saying almost nothing about Clofazimine which, according to your results, is much better. In the second part of the paper, in turn, no word about Arbidol. Please, somehow balance this, reflecting all the results that you have.
Technical details
-
In the abstract, “Arbidol and other Arbidol derivatives” - did you mean “Arbidol and its derivatives”? Please explain what derivatives? Also, two drugs in the beginning further substituted by three plus derivatives are rather confusing.
-
The ciphers 1 and 2 (Arbidol 1, and others below) are also confusing. Please delete them
-
Fig.2 is more suitable for materials and methods; Fig.3 is unnecessary at all - just reference would be enough.
-
Figs. 4, 5, and 7 should be made more readable
